# Contributing Factors to Perinatal Outcome in Pregnancies with Gestational Diabetes—What Matters Most? A Retrospective Analysis

**DOI:** 10.3390/jcm10020348

**Published:** 2021-01-18

**Authors:** Friederike Weschenfelder, Friederike Hein, Thomas Lehmann, Ekkehard Schleußner, Tanja Groten

**Affiliations:** 1Department of Obstetrics, University Hospital Jena, 07747 Jena, Germany; friederike.weschenfelder@med.uni-jena.de (F.W.); friederike.hein@hotmail.com (F.H.); ekkehard.schleussner@med.uni-jena.de (E.S.); 2Institute of Medical Statistics and Computer Science, University Hospital Jena, Friedrich Schiller University, 07747 Jena, Germany; thomas.lehmann@med.uni-jena.de

**Keywords:** gestational diabetes, obesity, gestational weight gain, perinatal outcomes, large for gestational age, C-section, NICU, body mass index

## Abstract

The aim of diabetes care of pregnant women with gestational diabetes mellitus (GDM) is to attain pregnancy outcomes including rates of large-for-gestational-age (LGA) newborns, pre-eclampsia, C-sections (CS) and other neonatal outcomes similar to those of the non-GDM pregnant population. Obesity and excessive weight gain during pregnancy have been shown to also impact perinatal outcome. Since GDM is frequently associated with elevated body mass index (BMI), we evaluated the impact of maternal prepregnancy BMI, development of GDM and gestational weight gain (GWG) during pregnancy on perinatal outcome. We compared 614 GDM patients with 5175 non-diabetic term deliveries who gave birth between 2012 and 2016. Multivariate regression analysis was used to evaluate the independent contribution of each factor on selected perinatal outcome variables. Additionally, subgroup analysis for obese (BMI ≥ 30 kg/m^2^) and non-obese women (BMI < 30 kg/m^2^) was performed. LGA was significantly influenced by BMI, GWG and GDM, while Neonatal Intensive Care Unit (NICU) admission was solely impacted by GDM. Maternal outcomes were not dependent on GDM but on GWG and prepregnancy BMI. These results remained significant in the non-obese subgroup only. Thus, GDM still affects perinatal outcomes and requires further improvement in diabetic care and patient counseling.

## 1. Introduction

Recently, awareness of gestational diabetes (GDM) and the necessity of treatment to reduce maternal and fetal complications has increased. Due to several changes in the diagnostic thresholds for GDM and in conjunction with the worldwide increase in non-communicable diseases, the incidence of GDM has risen over the last few decades. In 2017, the incidence of GDM in Germany was reported to be between 5.9% and 13.2%, respectively, which was probably even underestimated due to inconsistent recording [1,2]. The worldwide incidence of GDM is up to 11% using IADPSG (International Association of the Diabetes and Pregnancy Study Group) criteria [3].

GDM is known to be one of the major risk factors for large-for-gestational-age births (LGA) [4,5]. LGA infants are neonates that weigh at or above the 90th centile at birth adjusted to gender and gestational age at delivery. LGA deliveries are associated with prolonged labor, higher rates of cesarean sections (CS), shoulder dystocia and birth trauma [6,7]. LGA neonates are more likely to experience neonatal complications and to develop diabetes, obesity and metabolic syndrome as long-term consequences of developmental origins of adult disease hypothesis [8,9]. Additionally, the risk for CS in women with GDM was shown to be 30 to almost 100% higher than in women without GDM [10,11,12]. Furthermore, pregnancy-induced hypertension (PIH) and pre-eclampsia are known to be GDM-related pregnancy complications. Several study groups implicated an increased risk with ORs between 1.7 and nearly 3 [10,11,12]. The necessity to admit newborns of GDM mothers to Neonatal Intensive Care Unit (NICU) differs in the literature. Walters et al. could not find a difference between GDM and non-GDM patients, while Martin et al. showed a 2.5-fold increase in NICU admission [12,13]. Thus, the main goal of treating GDM patients is the reduction of GDM-associated pregnancy complications to common prevalence rates. To achieve this treatment goal, the German S3 guidelines for gestational diabetes published in 2011 even recommended quality checks of GDM treatment on a regular schedule [14]. Although detection rates have been improved with the implementation of glucose tolerance testing into regular pregnancy care and although treatment has been standardized to published guideline recommendations, the outcome of GDM pregnancies is still inferior to those not suffering from glucose intolerance.

Recently, prepregnancy body mass index (BMI) and gestational weight gain (GWG) have also been shown to be risk factors for LGA infants, elevated CS rates and impaired neonatal outcome in pregnancies with and without GDM [15,16,17,18,19,20,21]. Therefore, it is thought that, besides glycemic control, GWG within recommended limits would normalize the risk of LGA [22,23,24]. Since women with high GWG were more likely to undergo CS, Aiken et al. recently suggested controlling GWG to be one of the priorities in patient care following GDM diagnosis [25]. In patients with high prepregnancy BMI, the risk of NICU admission, LGA, pre-eclampsia and CS has also been shown to be directly affected by maternal obesity [26,27,28,29].

Accordingly, the aim of our study was to investigate the remaining impact of guideline conform monitored GDM on diabetes-related complications like LGA rates, necessity of NICU admission, pre-eclampsia and CS rate in a cohort of 5789 pregnancies (GDM n = 614; 10.6%). We hypothesized GDM to not further impact on these perinatal outcomes for the fetus and the mother if effects were adjusted for prepregnancy BMI and GWG. Furthermore, we wanted to analyze whether the expected independent effects of GDM, BMI and GWG differ depending on the presence of prepregnancy obesity (BMI ≥ 30 kg/m^2^) to find out what matters most.

## 2. Materials and Methods

### 2.1. Study Population

The final cohort (*n* = 5.789) of this study consist of two groups, derived from our clinical standardized perinatal database and the database of our outpatient department for diabetes and pregnancy: Non-GDM (*n* = 5.175) and GDM (*n* = 614). Mother infant-dyads of term deliveries (≥37 weeks of gestation) from 1st January 2012 until 31st December of 2016 were included. We excluded non-singleton pregnancies, preterm deliveries (<37 weeks of gestation), stillbirths, prenatal diagnosis of fetal malformations and cases with missing data concerning BMI and GWG. GDM diagnosis was based on the IADPSG and WHO-2013 criteria [30,31]. Diabetes care was provided according to the German S3 guidelines published in 2011 and provided by our hospital based outpatient department [14]. The non-GDM cohort was derived from term-deliveries in our university hospital during the same time-period without preexisting diabetes or GDM. (See Figure 1). Ethical approval was given by the local Ethical Committee of the Friedrich-Schiller-University, Jena, Germany (5280-09/17).

### 2.2. Study Outcomes and Measures

International Statistical Classification of Diseases and Related Health Problems (ICD-10) codes were used for identifying diabetes cases, as well as the documented risk numbers of the maternity record of the perinatal database (f.e. pre-eclampsia as O14.x).

BMI was calculated from maternal height and the prepregnancy weight. Concerning BMI all women were grouped into the Non-obese subgroup (BMI < 30 kg/m^2^ group) and the Obese subgroup women (BMI ≥ 30 kg/m^2^ group) according to the definitions of the world health organization [32]. We calculated GWG as the difference of the prepregnancy weight and the last documented weight during pregnancy. Excessive gestational weight gain (eGWG) was documented according to published IOM-criteria (Institute of Medicine) in case recommendations had been exceeded, according to the different BMI classes [33]. LGA and small-for-gestational-age (SGA) status were defined using Voigt’s percentiles for the body measurement of newborns and defined LGA above 90th percentile and SGA newborns below 10th percentile [34]. HbA1c levels of GDM mothers were measured on a regular four-weekly basis according to IFCC or NGSP/DCCT standard. Hb1c level at delivery was the last documented HbA1c level determined during pregnancy (target HbA1c < 5.7%; 39 mmol/mol [35]). Measurements up to a maximum of four weeks before delivery were included. Mean blood glucose (MBG) was calculated as the mean of all the patient’s 6-point self-monitored blood glucose profiles of the day prior to their regular consultations (target MBG 5.0–6.1 mmol/L).

Primary outcomes were the determination of impact factors on GDM-related perinatal complications: LGA status of the newborn and NICU admission representing neonatal complications (e.g., hypoglycemia and hyperbilirubinemia) as well as pre-eclampsia and CS rates representing maternal complications.

### 2.3. Statistical Analysis

Statistical analysis was performed with SPSS 24.0 (IBM Corp. Released 2016. IBM SPSS Statistics for Windows, Version 24.0. IBM Corp, Armonk, NY, USA). No prior sample size estimation was performed. A chi^2^ test or Fisher exact test was used to compare categorical data. Since most of the continuous data were not normally distributed, we used the median and interquartile range for data presentation and description. A Mann–Whitney U test was performed to compare continuous data between groups. Benjamini–Hochberg correction was used for controlling the familywise error rate due to multiple testing [36]. Adjusted odds ratios (ORs) for estimating the association of the GDM-related perinatal complications, GWG, BMI and GDM were determined using logistic regression. ORs with 95% confidence interval (CI) are presented. Generalized estimating equations were used to prove that there is no effect due to repeated observations because of multiple deliveries of one individual (*n* = 428; 7.4%). To address the issue of co-dependency of GDM, BMI and GWG, these factors and additionally the potential confounders (maternal age, weeks of gestation, parity and gender of the newborn) were included in the multivariate logistic regression model. The confounders are presented in the footnotes of the tables. A *p*-value < 0.05 was considered to indicate statistical significance (2-tailed).

## 3. Results

### 3.1. Patient Characteristics

The total cohort of this study included 5789 women: 5175 women without any type of diabetes (non-GDM) and 614 women with GDM. For cohort composition, see Figure 1. The descriptive data are shown in Table 1. Mean HbA1c levels at delivery was 5.5% (interquartile range (IQR) 5.2–5.7; 37 mmol/mol (IQR 33–39)) and mean blood glucose (5.8 mmol/l (IQR 5.6–6.1). GDM mothers had a significantly higher prepregnancy BMI (26.6 vs. 22.5 kg/m^2^), a higher frequency of overall obesity (BMI ≥ 30 kg/m^2^; 33.4% vs. 8%), were older (31 vs. 30 years), and showed higher rates of parity (1 vs. 0) but less GWG (12 vs. 15 kg). Rates of individuals with eGWG was similar in both groups (45.3% vs. 44.6%) (Table 1).

### 3.2. Perinatal Outcomes

Fetal birth weight was significantly higher in the GDM groups compared to the non-GDM group (3480 vs. 3420 g) despite the statistically gestational age (GA) difference with the higher GA in the non-GDM group. GDM pregnancies had a significantly higher frequency of LGA (12.8% vs. 7.6%) and SGA (8.7% vs. 6.1%) newborns. Pre-eclampsia (6.8 vs. 2.7%) and CS rate (29.6 vs. 22.5%) as well as NICU admissions (7.1 vs. 3.6%) were significantly higher in the GDM-group.

### 3.3. Independent Contributions of BMI, GWG and GDM on Perinatal Outcome

Prepregnancy BMI, GWG and GDM showed significant effects on LGA-rates after adjustment for maternal age, parity, gestational age and fetal sex. Each additional point in prepregnancy BMI increases the risk to deliver an LGA baby by 6% (odds ratio (OR) 1.06; 95% confidence interval (CI) 1.04–1.09), each additional kg GWG by 7% (OR 1.07; CI 1.05–1.10) and diagnosed GDM increased the risk by 45% (OR 1.45; CI 1.07–1.97) irrespective of all other confounders. eGWG (OR 1.31; CI 0.99–1.74) did not show a significant effect. GDM increased the risk for NICU admission by nearly 90% (OR 1.87; CI 1.23–2.84). Prepregnancy BMI (OR 1.01; CI 0.98–1.04), GWG (OR 0.97; CI 0.93–1.01 per kg) and eGWG (OR 1.39; CI 0.93–2.09) did not show a significant impact (Table 2) on NICU admission.

Adjusted analysis showed that GDM was without significant influence on pre-eclampsia rates (OR 1.35; CI 0.88–2.07), while prepregnancy BMI, GWG and eGWG had a significant effect. Each additional point in prepregnancy BMI increases the risk for pre-eclampsia by 15% (OR 1.15; CI 1.12–1.18). EGWG was highly associated with pre-eclampsia (OR 1.83; CI 1.18–2.86) and so was GWG (OR 1.05; CI 1.02–1.09 per kg). Multivariate analysis implicated only GWG and prepregnancy BMI as significant factors on CS rates, while GDM (OR 1.10; CI 0.90–1.36) and eGWG did not show an effect (OR 1.14; CI 0.94–1.37). Each additional point in prepregnancy BMI increases the risk for CS by 5% (OR 1.05; CI 1.04–1.07) and each additional kg GWG by 2% (OR 1.02; CI 1.01–1.04) (Table 2).

### 3.4. Subgroup Characteristics

Non-GDM and GDM mothers had significant different BMI values. For further analysis, we performed a subgroup comparison between non-obese with a prepregnancy BMI < 30 kg/m^2^ and obese mothers with a prepregnancy BMI ≥ 30 kg/m^2^. Subgroup characteristics are presented in Table 3.

In the non-obese subgroup analysis, LGA rates differed significantly between GDM and non-GDM women (13 vs. 7.3%), so did birth weight (3465 vs. 3420 g), NICU admission (8.4 vs. 3.4%) and CS rates (29.3 vs. 21.5%). In contrast, subgroup analyses of obese mothers revealed no difference concerning LGA rates (GDM 12.4% vs. non-GDM 11.3%), 5 min-APGAR (9 vs. 9), birth weight (3520 vs. 3500 g), NICU admission (4.3 vs. 6%), pre-eclampsia (12.6 vs. 8.9%) and CS rate (30.6 vs. 33.3%).

Furthermore, individuals with eGWG were significantly more common among obese non-GDM patients (63.8 vs. 51.5%), while, in the non-obese subgroup, both GDM and non-GDM individuals showed excessive weight gain in ~43% of the cases. Absolute GWG was significantly lower in both GDM subgroups, whether obese or non-obese, compared to non-GDM mothers as shown in Table 3 (BMI < 30 kg/m^2^ subgroup: non-GDM 15 vs. GDM 13.7 kg and BMI ≥ 30 kg/m^2^: non-GDM 12 vs. GDM 9.3 kg).

### 3.5. Contributing Factors in Subgroups

We performed the equivalent multivariate analysis using generalized estimating equations for both subgroups as we did for the entire cohort. ORs are presented in Figure 2. In the non-obese subgroup, the results did not change compared to the entire cohort analysis: GWG, prepregnancy BMI and GDM showed a significant impact on LGA status: each additional kg weight gain goes along with an 8% increase in LGA rate (OR 1.08; CI 1.05–1.10), every additional extra BMI point with an increase of 12% (OR 1.12; CI 1.08–1.17) and GDM with 67% (OR 1.67; CI 1.18–2.36). Non-obese women with GDM had a 2.6-fold higher risk for NICU admission (OR 2.60; CI 1.70–3.92). BMI, GWG and eGWG had no effect on NICU admission. Risk for pre-eclampsia was significantly increased by 14% per BMI point (OR 1.14; CI 1.07–1.22), by 8% per kg GWG (OR 1.08; CI 1.03–1.12) and doubled in case of eGWG (OR 2.11; CI 1.22–3.66). Risk for CS in non-obese was significantly increased by GWG (OR 1.02; CI 1.01–1.04) and BMI (OR 1.06; CI 1.03–1.08). GDM showed no independent effect on CS and pre-eclampsia in this subgroup.

Multivariate analysis of the obese subgroup showed GDM to be without impact on rates of LGA, NICU admission, pre-eclampsia or CS (see Figure 2)

In the obese subgroup, none of the determinants showed an independent increase in LGA. In the obese group, NICU admission was only impacted by GWG (OR 0.93; CI 0.86–1.0). BMI showed to be an individual factor impacting on the risk to develop pre-eclampsia. The risk for pre-eclampsia increased by 12% with each additional BMI point (OR 1.12; CI 1.05–1.19). None of the parameters was solely associated with CS rate in obese mothers.

## 4. Discussion

Despite many investigations and studies, it is still not clear why there are major differences in the perinatal outcomes of women without GDM and women with diagnosed GDM despite standardized diabetes care according to international guidelines. Accordingly, in this study, closely monitored GDM still had a significant impact on adverse perinatal outcomes. Hba1c levels and MBG both showed that metabolic control could still be improved. HbA1c at delivery with a 75% IQR of 5.7% (39 mmol/mol) means that 25% of the HbA1c values would still be above the target HbA1c of 5.7%. Women with GDM were more likely to have higher rates of LGA infants, NICU admissions, pre-eclampsia and CSs compared to the non-GDM cohort. In contrast to our findings, Kleinwechter et al. recently published a quality analysis of 1074 singleton pregnancies that showed an acceptable concordance of the outcomes of GDM pregnancies compared to the general obstetric background population from 2012–2017 in Germany. They could not find different rates of LGA (7.2 vs. 9.2%), NICU admission (13.2 vs. 11.1%) or emergency CSs (16.5 vs. 15.7%, all type of CSs 34.8 vs. 27.6%). However, in our study, GDM patients have been compared to a cohort of non-GDM low risk term deliveries and rates of NICU admission (3.6%), preeclampsia (2.7%), CS (22.5%) and LGA (7.6%) were profoundly lower in our non-GDM collective. Thus, inclusion of diabetic pregnancies, as well as other risk pregnancies, in the comparison group of Kleinwechter et al. might well account for the different results [37].

Nevertheless, there are previous studies that refuted the groundbreaking effects of GDM treatment on the perinatal outcomes such as we did [10,11]. The remaining question is why it seems not to be possible to normalize pregnancy outcome in GDM pregnancies.

Notably, prepregnancy BMI and GWG, besides parity and maternal age revealed to be significantly different in GDM patients compared to the non-GDM group. Consequently, we performed adjusted analysis and showed that, besides GDM, prepregnancy BMI and GWG significantly affected perinatal outcome, leaving GDM without impact on pre-eclampsia and CS rates. This matches the results of Bianchi et al. where GDM also did not show an independent effect on CS rates [38]. This is contrary to previous findings where GDM profoundly impacted on CSs rates and pre-eclampsia [10,11,12]. However, comparison of CS rates is in general hampered by the wide ranges of international CS rates from 4% in West and Central Africa to 44% in Latin America and the Caribbean [39].

Furthermore, differences in the outcomes documented for GDM pregnancies might also be due to the heterogeneity of pathophysiological reasons for GDM. Liu et al. performed a study to investigate the impacts of the different GDM subtypes on perinatal outcome assuming differences depending on the cause of GDM diagnosis (insulin resistance vs. beta cell dysfunction vs. mixed types). According to their findings, GDM patients with both above-mentioned traits had significantly higher risks for LGA or any adverse outcomes. Nevertheless, these significant results vanished after adjusting for maternal age and prepregnancy BMI, leaving none of the expected individual risk factors for an adverse outcome to be independent in this study [40]. Therefore, adjusting for major impact factors such as maternal age, parity, gestational age and sex of the offspring became an important issue in our study—still leaving individual risk factors for the defined perinatal outcomes. This is in accordance with the accepted fact that neonatal outcome is predominantly influenced by the glycemic control during pregnancy and rather less by the type of Diabetes. However, maternal outcomes were still affected by prepregnancy BMI and GWG and neonatal outcomes like LGA rate and NICU admission were still impacted by GDM. The risk for LGA was increased by 45% and for NICU admission almost doubled in GDM. However, adjusted analysis revealed, NICU admission to be solely influenced by the diagnosis of GDM, while the risk of LGA was also significantly increased by prepregnancy BMI (OR 1.06) and per kg weight gain (OR 1.07). Previous studies focusing on LGA confirm our findings concerning the impacts of BMI, GWG and GDM on LGA rates [19,22,23,24,25,41].

Since prepregnancy BMI plays such a major role in effecting perinatal outcomes, we performed a subgroup analysis comparing obese and non-obese pregnancies. Surprisingly, in obese mothers with or without GDM, none of the shown differences of maternal or neonatal complications persists (see Table 3). These results are confirmed by logistic regression analysis showing that individual impacts on maternal and neonatal complications changed profoundly (see Figure 2). In particular, the effect of GDM on neonatal outcome remained significant in the non-obese group only, raising the risk for LGA by 67% and for NICU admission by 2.5-fold. Additionally, GWG was shown to be an independent risk factor for LGA in the non-obese mothers. Contrarily, for the obese subgroup no relevant effector could be determined further affecting outcome beyond the fact of being obese. These results match our earlier findings published in 2019 [24].

CS rate was not further influenced by BMI and GWG in the obese group, whereas they remained to be of significant impact in the non-obese subgroup. Aiken et al. showed that higher total gestational weight gain was related to higher CS rates [25] but the authors did not discriminate between BMI classes. Miao et al. performed multivariate analysis and showed that CS rates in GDM pregnancies were significantly higher in overweight and obese pregnancies adjusted for GWG. However, they did not include GDM as an impact factor in their analysis [19].

Pre-eclampsia was impacted by prepregnancy BMI in both groups, whereas the significant effect of GWG and excessive GWG could, again, only be verified for the non-obese subgroup. Consistently, maternal outcomes were not influenced by GDM in both groups. Our data confirm BMI as a well-known risk factor for pre-eclampsia [28,29]. We also agree with study of Hutcheon et al. that demonstrate excessive GWG to be a stronger risk factor for pre-eclampsia in lean than in obese women [42]. The observed individual impact of GWG on pre-eclampsia was less in the obese subgroup. This finding is likely consequent to the preexisting predominantly inflammatory milieu that drives the obese into a pre-eclamptic state [43].

Our data show that GWG was significantly less in the GDM cohort (12 vs. 15 kg) where patients were counselled for diet control and weight gain aims as recommended in the IOM guidelines for each BMI group. Nevertheless, both groups showed an excessive GWG of 45% that is consistent with the findings of Goldstein et al. (47%) and Rogozińska et al. (37%) [44,45]. Aiken et al. proved that counselling concerning GWG even after GDM diagnosis had a positive influence on improving outcomes such as LGA [25]. Our data clearly show that especially non-obese women would benefit from controlling GWG to prevent adverse outcomes and should be counselled early in pregnancy. Strikingly, we could not find a relevant individual impact on adverse outcome parameters in the obese groups, leaving the prepregnancy BMI to be particularly relevant. That clearly underscores the importance of informing obese women in their reproductive years about pre-gestational weight loss and the long-lasting benefits for them and their future children.

The meta-analysis of the LifeCycle Project-Maternal Obesity and Childhood Outcomes Study Group showed that GWG had a significant influence on perinatal outcome as well, but they also emphasized that available gestational weight gain ranges showed limited predictive value for the outcomes assessed only. They recommend using estimates for optimal gestational weight gain for counseling [46].

### Strength and Limitations

A strength of our study is the comparison of a large number of closely monitored well-documented GDM pregnancies with a large number of healthy mother–infant dyads receiving the same level of health care during their peripartum period, minimizing potential bias due to different perinatal procedures.

There are some limitations of the present study that might include the unicentric, retrospective design and the reliability of the electronic records. Another confounder might be the rather high percentage of insulin-treated pregnancies (43%). Due to frequent admission of complicated GDM cases to our specialized unit, the rate of insulin-treated pregnancies is relatively high compared to the average of about 30% documented in the German GDM register (GestDiab) [47].

Additionally, in our multivariate analysis, other influencing factors (e.g., migration status, smoking, time of GDM diagnosis, interpregnancy weight gain) were not taken into account due to limited data availability. However, for ethnicity, it has to be stated that, in the Jena obstetric population, the proportion of non-Caucasian women is far below 10%.

## 5. Conclusions

GDM was determined to be a significant risk factor for impaired neonatal outcome in the non-obese patients. Consistent with previous findings, GWG revealed to be a profound strong but influenceable factor on neonatal and maternal outcome. Especially non-obese mothers benefit from less weight gain during pregnancy, irrespective of their diabetic status concerning adverse outcomes. We suggest that all women should be counseled properly on gestational weight gain and nutrition during pregnancy.

## Figures and Tables

**Figure 1 jcm-10-00348-f001:**
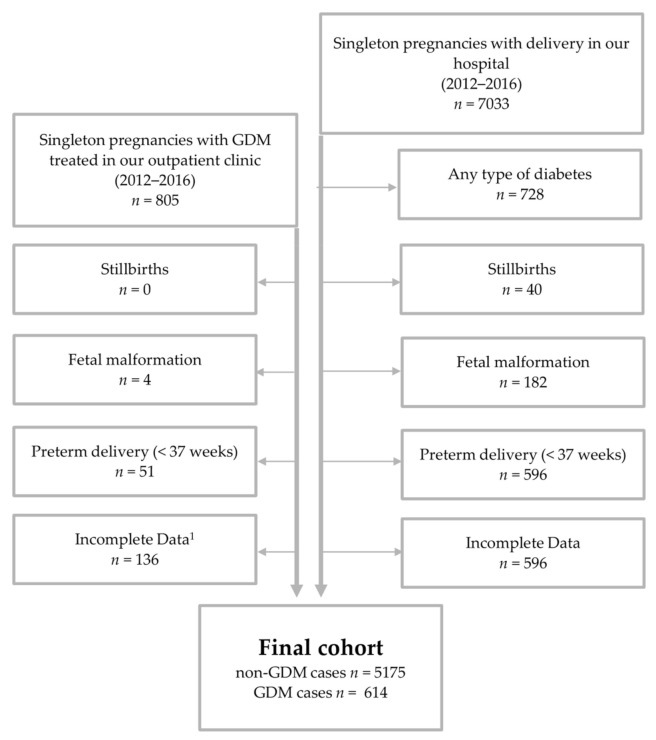
Cohort composition: The final cohort consists out of 614 gestational diabetes (GDM) cases and 5175 non-GDM term deliveries. The non-GDM cohort was derived from initially 7033 singleton mother infant dyads from 2012 until 2016. All cases of stillbirths, fetal malformations, preterm deliveries (<37 weeks) and cases with incomplete data were excluded in both groups. All types of diabetes cases were excluded in the non-GDM group. ^1^ Higher rate of exclusions due to missing information about deliveries ex muros.

**Figure 2 jcm-10-00348-f002:**
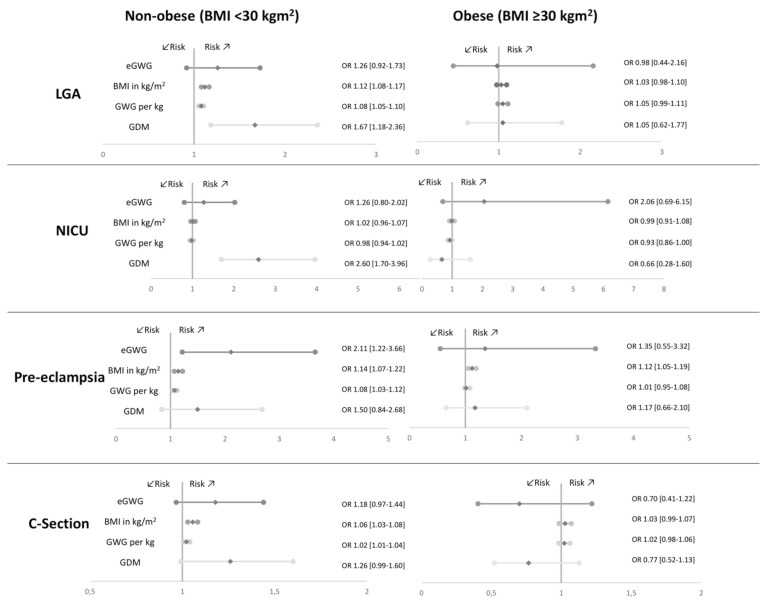
Adjusted odds ratios (ORs) of perinatal complications for the non-obese and obese subgroups: large for gestational age (LGA), neonatal intensive care unit (NICU) admission, pre-eclampsia and C-Section. Results from the logistic regression models adjusted for maternal age, parity, gestational age and sex of the newborn are presented: excessive gestational weight gain (eGWG), body mass index (BMI), gestational weight gain (GWG) per kg and gestational diabetes (GDM).

**Table 1 jcm-10-00348-t001:** Descriptive parameters of pregnancies with and without GDM.

Entire Cohort (*n* = 5789)	Non-GDM (*n* = 5175)	GDM (*n* = 614)	*p*
Maternal age (years)	30 (27–33)	31 (28–35)	<0.001 *
Parity	0 (0–0)	1 (0–1)	<0.001 *
Prepregnancy BMI (kg/m^2^)	22.5 (20.5–25.3)	26.6 (23.0–32.2)	<0.001 *
BMI categories	281 (5.4)	5 (0.8)	<0.001 *
BMI < 18.5 kg/m^2^ (underweight)	281 (5.4)	5 (0.8)	
BMI 18.5–24.9 kg/m^2^ (normal weight)	3514 (67.9)	246 (40.1)	
BMI 25–29.9 kg/m^2^ (overweight)	965 (18.6)	158 (25.7)	
BMI 30–34.9 kg/m^2^ (obesity class I)	291 (5.6)	105 (17.1)	
BMI 35–39.9 kg/m^2^ (obesity class II)	87 (1.7)	68 (11.1)	
BMI ≥ 40 kg/m^2^ (obesity class III)	37 (0.7)	32 (5.2)	
GWG (kg)	15 (11–18)	12 (8.5–16.6)	<0.001 *
Excessive GWG	2345 (45.3)	280 (45.6)	0.89
MBG (mmol/l)	-	5.8 (5.6–6.1)	
HbA1c at delivery in %	-	5.5 (5.2–5.7)	-
HbA1c at delivery in mmol/mol	-	37 (33–39)	-
Insulin treatment	-	264 (43)	-
GA at delivery (weeks)	39 (39–40)	39 (38–40)	<0.001 *
Birth weight (g)	3420 (3140–3720)	3480 (3193–3790)	0.002 *
LGA	395 (7.6)	77 (12.8)	<0.001 *
SGA	451 (8.7)	37 (6.1)	0.03 *
APGAR 5	9 (9–10)	9 (9-10)	0.278
NICU	188 (3.6)	40 (7.1)	<0.001 *
Pre-eclampsia	142 (2.7)	35 (6.8)	<0.001 *
Spontaneous Delivery	3606 (69.7)	403 (65.6)	0.042 *
C-section (planned and emergency)	1164 (22.5)	182 (29.6)	<0.001 *

Data are n (%) or median (interquartile range) unless otherwise specified. * remaining significant (*p* < 0.05) after using Benjamini–Hochberg correction for multiple testing; GWG: gestational weight gain; GA: gestational age; LGA: large for gestational age, defined as birth weight >90th percentile; SGA: small for gestational age, defined as birth weight <10th percentile; NICU: neonatal intensive care unit; MBG: mean blood glucose.

**Table 2 jcm-10-00348-t002:** Adjusted odds ratios (ORs) for perinatal complications.

	LGA ^a^		NICU ^b^		Pre-Eclampsia ^c^		CS ^d^	
	OR (CI)	*p*	OR (CI)	*p*	OR (CI)	*p*	OR (CI)	*p*
eGWG	1.31 (0.99–1.74)	0.06	1.39 (0.93–2.09)	0.11	1.83 * (1.18–2.86)	<0.01	1.14 (0.94–1.37)	0.18
Prepregnancy BMI (kg/m^2^)	1.06 * (1.04–1.09)	<0.01	1.01 (0.98–1.04)	0.71	1.15 * (1.12–1.18)	<0.01	1.05 * (1.04–1.07)	<0.01
GWG (per kg)	1.07 * (1.05–1.10)	<0.01	0.97 (0.93–1.01)	0.09	1.05 * (1.02–1.09)	<0.01	1.02 *(1.01–1.04)	0.01
GDM	1.45 * (1.07–1.96)	0.02	1.87 * (1.23–2.84)	<0.01	1.35 (0.88–2.07)	0.18	1.10 (0.90–1.36)	0.36

Adjustments were made for maternal age, parity, gestational age and sex of newborn. CS: cesarean section; LGA: large for gestational age, defined as birth weight > 90th percentile; GDM: gestational diabetes mellitus; eGWG: excessive gestational weight gain; NICU: neonatal intensive care unit; * *p* < 0.05; number of included cases in Modell: ^a^
*n* = 5770; ^b^
*n* = 5735; ^c^
*n* = 5684; ^d^
*n* = 5773.

**Table 3 jcm-10-00348-t003:** Descriptive parameters of pregnancies with and without GDM in the non-obese (BMI < 30 kg/m^2^) or obese (BMI 30 ≥ kg/m^2^) subgroups.

	Non-Obese BMI < 30 kg/m^2^ ( *n* = 5166)			Obese BMI ≥ 30 kg/m^2^ ( *n* = 623)		
	Non-GDM (*n* = 4758)	GDM (*n* = 408)	*p* ^#^	Non-GDM (*n* = 417)	GDM (*n* = 206)	*p* ^#^
Maternal age (years)	30 (27–33)	31 (28–35)	<0.001 *	30 (26–33)	31 (27–34)	0.027 ^‡^
Parity	0 (0–1)	1 (0–1)	0.003 *	1 (0–1)	1 (0–2)	0.24
Prepregnancy BMI (kg/m^2^)	22.1 (20.3–24.4)	23.9 (21.7–26.6)	<0.001 *	32.8 (31.1–35.5)	34.8 (32.2–37.7)	<0.001 *
GWG (kg)	15 (12–18)	13.7 (9.8–17.3)	<0.001 *	12 (7–16)	9.3 (5–14)	<0.001 *
eGWG	2079 (43.7)	174 (42.6)	0.716	266 (63.8)	106 (51.5)	0.004 *
HbA1c at time of delivery (%)	-	5.5 (5.2–5.7)	-	-	5.5 (5.3–5.7)	-
HbA1c at time of delivery (mmol/mol)	-	37 (33–39)			37 (34–39)	-
Insulin treatment (%)	-	143 (35)	-	-	121 (58.7)	-
GA at delivery (weeks)	40 (39–40)	39 (39–40)	0.004 *	39 (39–40)	39 (38–40)	0.001 *
Birth weight (g)	3420 (3130–3705)	3465 (3190–3790)	0.026 *	3500 (3175–3830)	3520 (3220–3796)	0.735
LGA	348 (7.3)	52 (13.0)	<0.001 *	47 (11.3)	25 (12.4)	0.690
SGA	421 (8.9)	27 (6.7)	0.166	30 (7.2)	10 (5.0)	0.383
APGAR 5	9 (9–10)	9 (9–10)	0.327	9 (9–10)	9 (9–10)	0.684
NICU	163 (3.4)	32 (8.4)	<0.001 *	25 (6.0)	8 (4.3)	0.560
Pre-eclampsia	105 (2.2)	14 (4.1)	0.039 ^†^	37 (8.9)	21 (12.6)	0.220
C-section	1025 (21.5)	119 (29.3)	0.001 *	139 (33.3)	63 (30.6)	0.525

Data are *n* (%) or median and interquartile range unless otherwise specified; ^#^ comparison GDM vs. non-GDM in either the non-obese or obese subgroup; * remaining significant (*p* < 0.05) after using Benjamini–Hochberg correction for multiple testing; ^†^ not significant (*p* > 0.05) after correction in the non-obese group: pre-eclampsia *p* = 0.51; ^‡^ not significant (*p* > 0.05) after correction in the obese-group: maternal age *p* = 0.07. eGWG: excessive gestational weight gain; GWG: gestational weight gain; GA: gestational age; LGA: large for gestational age, defined as birth weight >90th percentile; SGA: small for gestational age, defined as birth weight <10th percentile; NICU: neonatal intensive care unit.

## Data Availability

The data presented in this study are available on reasonable request from the corresponding author.

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
