# Peer review of "Contributing Factors to Perinatal Outcome in Pregnancies with Gestational Diabetes—What Matters Most? A Retrospective Analysis"

_jcm, 2021, doi:10.3390/jcm10020348_

Round 1

Reviewer 1 Report

The authors present a retrospective cohort study designed to evaluate the independent effect of gestational diabetes (GDM), body mass index (BMI) and gestational weight gain (GWG) on maternal and perinatal outcomes known to be associated with GDM. Pregnant women with GDM are compared to pregnant women without GDM cared for at a university hospital over a 5 year period using an institutional perinatal database and an outpatient diabetes and pregnancy database. Multivariate logistic regression analysis was applied to the whole cohort and a subgroup analysis of obese and non-obese women was also performed. GDM, BMI and GWG were significantly associated with large for gestational age (LGA) newborns and GDM was associated with NICU admission. The authors conclude that while maternal complications such as preeclampsia and cesarean delivery rate appear to be influenced by BMI and GWG, neonatal outcomes such as LGA and NICU admission appear to be independently influenced by GDM, underscoring the importance of identification and optimal diabetes and pregnancy management of women with GDM.

Comments:

1) This study is relevant to current clinical practice and provides interesting and informative results from a large single centre cohort using appropriate data analysis methods. Overall, the manuscript is well written and the methods and results are clearly presented. The authors' conclusions are supported by their findings.

2) I would propose that the opening sentence of the Abstract could be improved upon. May I suggest " The aim of diabetes care of pregnant women with gestational diabetes mellitus (GDM) is to attain pregnancy outcomes including rates of large for gestational age (LGA) newborns, pre-eclampsia, C-sections (CS) and other neonatal outcomes similar to those of the non-GDM pregnant population."

3) I would suggest that the exclusions (lines 98-101) be moved to Methods subsection 2.1 Study Population since exclusions define the study population and would then appropriately precede Figure 1.

3) The authors state that no prior sample size calculation was performed. Without a sample size calculation to inform the number of GDM and non-GDM pregnancies required to provide sufficient power to establish a significant difference between groups in a selected primary or composite outcome, can we be confident with the results given the subgroup analysis and multiple comparisons?

4) The potential co-dependency of GDM and BMI (and possible GWG) are not addressed in the Statistical methods or Discussion sections of the paper. Though not clearly stated in 2.3 Statistical Analysis subsection, one assumes that all three factors were included in the logistic regression model along with the stated potential confounders. I would suggest that this aspect of regression modelling, and any other statistical techniques used to control for codependency of the primary factors of interest in this study, be clearly described to strengthen the findings of "independent contribution" of GDM,BMI and GWG. I would also suggest that the issue of potential codependence of these factors and the author's confidence in in independent contributions to the pregnancy outcomes evaluated be addressed in the Discussion section.  

Reviewer 2 Report

This is a well designed, clearly presented and nicely written pape that adds to the current body of evidence.

I have only one methodological concern that I feel is important to resolve. Can you please detail how (the method employed) the women's weight and height was measured? Were these parameters measured by the attending (ie not self reported)? For example, weigth was measured in kilograms using eg Tanita® scale & height in meters using a Seca® wall-mounted stadiometer according to departmental procedures.

Discusion - may benefit from the addition of a line or 2 about the implications of viseral fat vis a vis obesity per se?

Minor comments:

Can you please include HbA1c values in mmol/mol alongside values in %

e.g., Line 113 HbA1c <5.7% (39mmol/mol)

Spell Check

Line 130 proof change to prove

Line 288 und to and

Line 320 that match others studies results with 47% .....

change to

that is consitent with the findings of Goldstein et al (47%) and Rogozinska et al (66.7%) [44,45] 

Line 312 suggest remove "the" before BMI

Line 313  Hutcheon et al that demonstrates excessive GWG to be

Lines 315 - a little confusing  - consider revising - suggest 

The observed individual impact of GWG on pre-eclampsia was less in  the obese subgroup. This finding is likely consequent to the preexisting predominantly inflammatory milieu that drives the obese into a pre-eclamptic state [43]. 

Line 324 suggest remove "already"

Line 352

Consistently to Consistent with   GWG was revealed ...
